# Artificial Intelligence in Surgical Learning

**Niklas Pakkasjärvi** [1,*] , **Tanvi Luthra** [2] **and Sachit Anand** [2]

1   Department of Pediatric Surgery, Turku University Hospital, 20521 Turku, Finland
2   Department of Pediatric Surgery, All India Institute of Medical Sciences, New Delhi 110029, India
*   Correspondence: niklas.pakkasjarvi@tyks.fi; Tel.: +358-02-3130000

**Abstract:** (1) Background: Artificial Intelligence (AI) is transforming healthcare on all levels. While AI shows immense potential, the clinical implementation is lagging. We present a concise review of AI in surgical learning; (2) Methods: A non-systematic review of AI in surgical learning of the literature in English is provided; (3) Results: AI shows utility for all components of surgical competence within surgical learning. AI presents with great potential within robotic surgery specifically; (4) Conclusions: Technology will evolve in ways currently unimaginable, presenting us with novel applications of AI and derivatives thereof. Surgeons must be open to new modes of learning to be able to implement all evidence-based applications of AI in the future. Systematic analyses of AI in surgical learning are needed.

**Keywords:** artificial intelligence; surgical learning; machine learning; minimally invasive surgery

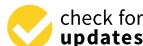



## 1. Introduction

Artificial intelligence (AI) is transforming healthcare [1,2]. AI can improve patient care by analyzing large amounts of data to help make more informed decisions regarding treatments and enhance medical research through analyzing and interpreting data from clinical trials and research projects to identify subtle but meaningful trends beyond ordinary perception. AI refers to the simulation of human intelligence in computers, where AI systems can perform tasks that require human-like intelligence like speech recognition, visual perception, pattern-recognition, decision-making, and language processing. AI has several subdivisions, including machine learning, natural language processing, computer vision, and robotics. By automating specific routine tasks, AI can improve healthcare efficiency. By leveraging machine learning algorithms, AI systems can offer new opportunities for enhancing both the efficiency and effectiveness of surgical procedures, particularly regarding training of minimally-invasive surgery. As AI continues to advance, it is likely to play an increasingly significant role in the field of surgical learning.

The use of AI in surgical learning has the potential to transform the way in which surgeons are trained. Surgical training has undergone significant changes in recent years through the introduction of simulation and task-based training. AI complements this path with great promise. However, while AI shows immense potential, the clinical implementation is lacking, including the inclusion of AI into formal medical curricula [3]. AI is especially useful for the creating of simulations of surgical procedures, thereby allowing trainees to practice skills in a controlled environment and to develop a better understanding of the complex task of surgery [4,5]. Personalized training materials can be created with the aid of AI, tailoring content to individual needs and learning styles, thus providing a more efficient and effective means of learning, especially considering how medical education models have previously relied on experimental evidence guiding printed textbooks, involving a slow process outpaced by current society [6]. During surgical procedures, AI can be utilized to provide real-time guidance, helping in decision-making and minimizing the risk of complications. AI can be taught in decision-making through its ability to analyze vast datasets from past surgeries, identifying patterns and trends to best improve modes

of action and thereby subsequent outcomes. AI can assist in diagnostics and treatment planning for surgical patients, helping surgeons to identify the best stratified treatment based on individual circumstances. This way, surgeons can make better-informed decisions and improve patient care. In the future, AI can be used for controlling robotic surgical instruments, allowing for more widespread application of minimally invasive surgery. As surgeons in training begin to integrate the use of AI into their learning, it is important that they have a clear understanding of both the possibilities and limitations of this technology. While AI has the potential to enhance traditional teaching methods, it is important to recognize that it will never completely replace the role of human instructors. It will serve as a supplement to traditional teaching methods, helping to improve the learning experiences for surgeons in training, but the human instructor is and remains crucial for the training process. They should be the ones who guide, evaluate, and give feedback to the trainee. It is important that surgeons in training are able to understand the potential of AI to assist in their learning, but also recognize its limitations and the ongoing importance of human instruction.

The use of AI in surgical learning has the potential to enhance the quality and efficiency of training, thus improving clinical results. As the field of surgery and technology continues to advance, it is highly likely that the integration of AI into the surgical training process will become increasingly prevalent. This integration will likely take place in ways that are currently difficult to anticipate or comprehend as the capabilities and possibilities of AI continue to evolve. The use of AI in surgical training has the potential to revolutionize the way in which surgeons are trained and could bring about significant improvements in the overall quality of surgical care. However, it is also important to consider the potential ethical and practical issues that may arise with the increasing use of AI in this field. Through the integration of AI into surgical training, surgical learners can acquire surgical competence in improved ways to further improve patient care. This integration also requires the surgeon, ethicists and researchers to work together to ensure the safety, fairness and effectiveness of the technology. While AI shows great potential, it is not without limitations and, despite great promise, clinical application is thus far scarce. We must continue to monitor, evaluate and adapt to the increasing integration of AI into surgical training. The rapid and near exponential growth in publications regarding AI poses challenges for clinicians to stay updated. This rapid review provides a concise outlook of the current status of different applications of AI in surgical learning.

## 2. Materials and Methods

We conducted a non-systematic rapid literature review. We searched PubMed on 18 December 2022 for articles on 'artificial intelligence', 'machine learning', and 'surgical learning'. Snowballing was implemented for further studies of relevance. Only articles written in English and available for open access were included. Data were extracted manually in several increments, starting with title and abstract scanning, proceeding to text review. AI (OpenAI GPT3) was applied for literature analysis in select situations for summarizing data from the included texts. AI was also used for spelling and grammar corrections. All studies of relevance to AI and surgical learning were considered eligible for initial analysis. Final analysis was done by the authors in person.

## 3. Results of Current Status of AI in Surgical Learning

During recent years, there has been a significant expansion in the number of studies on artificial intelligence in surgery (Figure 1). AI has the potential to revolutionize surgical education and training. Its potential applications in surgical learning are numerous, but there is still a lack of evidence demonstrating its utility in a clinical teaching setting. Despite this, AI has shown particular promise in the training of minimally-invasive surgery, as well as in aiding in diagnosis and decision-making processes. The use of AI in these areas has the potential to significantly improve patient outcomes and increase the efficiency of surgical procedures. The utilization of AI in surgical education is an area of ongoing

expansion, with its potential applications only constrained by the creativity and ingenuity of those who employ it. An overview of the present utilization of AI in surgical learning, including accompanying limitations and restrictions, is provided below (Figure 2).

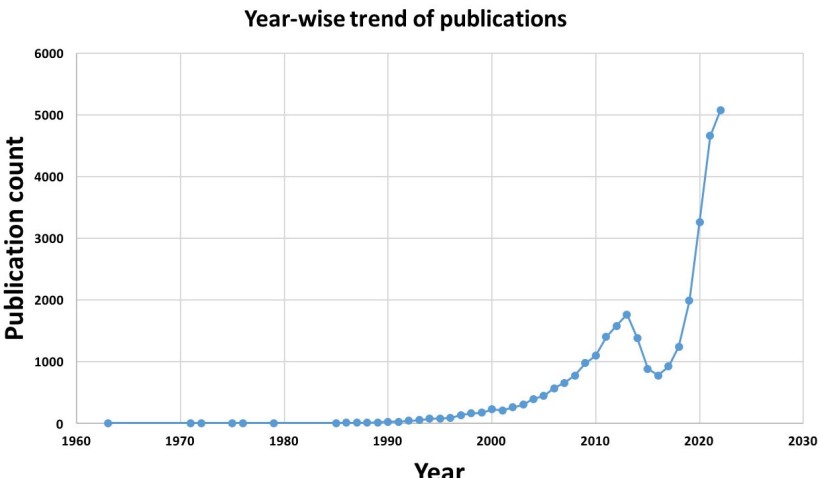

**Figure 1.** Cumulation of studies on AI in surgery during recent years in PubMed.

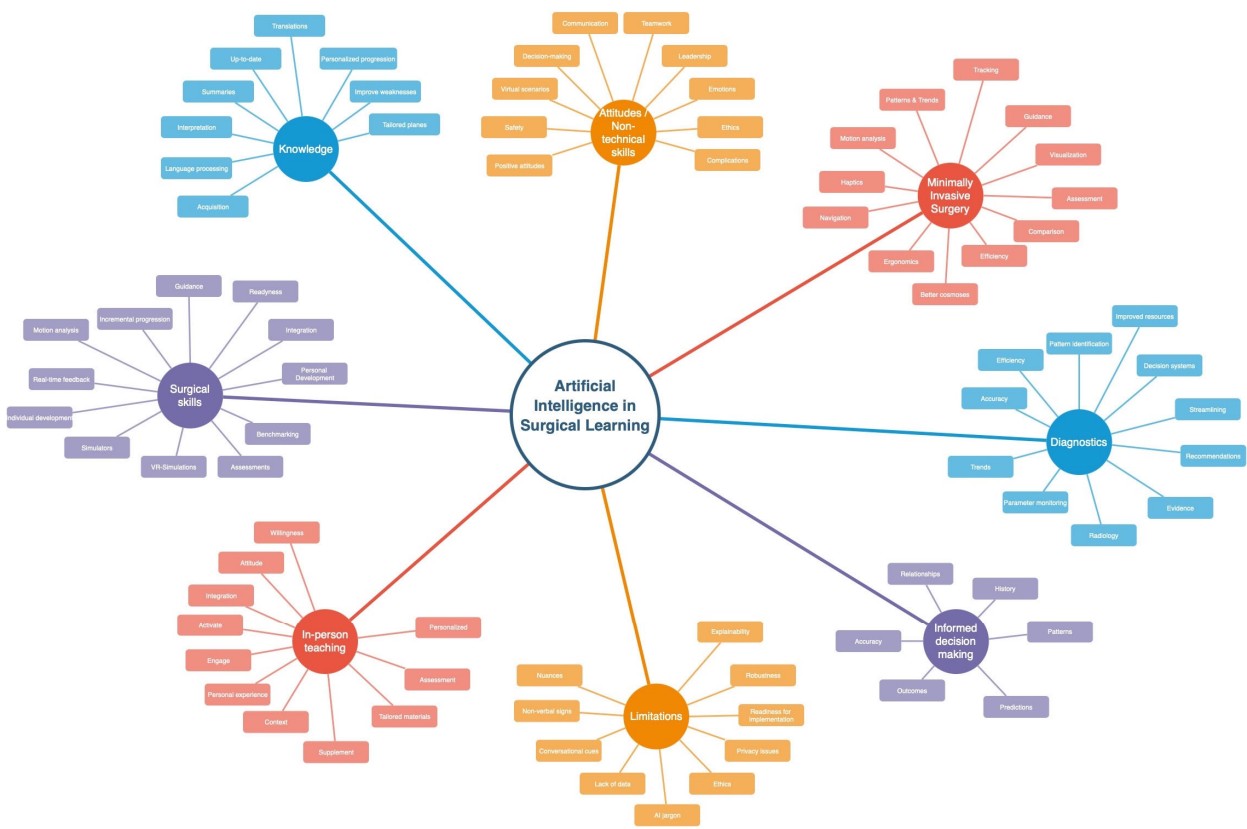

**Figure 2.** Applications and limitations of AI in surgical learning.

## 3.1. AI in Surgical Learning

### 3.1.1. AI in Learning Surgical Competence

Surgical competence encompasses a wide range of abilities. Traditionally, competence is thought to entail the components of knowledge, skills, and attitudes [7]. AI can potentially play a significant role in the development and improvement of all components of surgical competence, including surgical skills, and shorten learning curves of surgical

procedures [8]. AI is especially suited to simulation-based training [9]. Simulation-based training has become essential to surgical education, improving both confidence and performance of trainees. By using virtual reality or other simulation technologies, AI can provide a realistic and safe environment for surgical residents and other trainees to practice and hone their technical skills. Simulation-based training allows trainees to gain experience and confidence in performing procedures without the risk of harm to actual patients [10]. Simulation training can also help the trainee to retain the skills and use them effectively in real-world situations. AI can further provide real-time feedback and assessment of performance, allowing for more personalized and efficient skill development [11]. Through motion-based analysis, AI has proven utility to analyze the hand movements and provide guidance on how to improve technique, and it can highlight areas of the procedure that may require additional practice [12–14]. In addition to improving technical skills, simulation-based training with AI can also help surgeons develop non-technical skills such as crisis management and decision-making under pressure [15]. In clinical decision-making, AI could support by accelerating the process and helping doctors to understand semantics of language [15]. AI can also help in the evaluation of competences and readiness for independent practice [16–18]. While the use of objective methods for assessing technical surgical tasks has been shown to be valid through past research, the integration of AI-based feedback systems into these assessments may bring about new challenges. These challenges may arise in a wide range of applications, highlighting the need for further exploration and examination in order to fully understand the potential limitations and opportunities presented by the use of AI in this field.

The acquisition of surgical knowledge can be facilitated by AI [19]. AI-powered tools such as natural language processing can be used to extract and analyze large amounts of data from medical literature and other sources, providing a comprehensive overview of current best practices and evidence-based approaches to surgical procedures [20]. This can be especially useful for trainees who are trying to keep up with the rapidly evolving field of surgery and stay current with the latest guidelines and recommendations. AI-based language processing tools can aid in both summarizing and explaining complex literature in personal ways to suit individual needs and language requirements. In addition, AI can assist in the creation of personalized study plans and learning materials based on individual trainee needs and knowledge gaps. With AI, a trainee's performance on practice exams can be analyzed to help identify specific topics of incompetence, generating tailored study materials to help them improve in those areas. Through the application of AI to analyze learning, teachers can focus resources from manual analysis to teaching weaknesses on an individual basis. The rapid evolution of AI-based applications forces teachers to maintain awareness of current trends in AI while simultaneously adapting their teaching to novel methods.

The development of positive attitudes is an important aspect of surgical training, as they can impact patient care and safety [21–23]. AI can enhance attitude training through virtual patient cases and scenarios that allow trainees to practice and develop their communication and decision-making skills, as well as their ability to handle complex ethical situations [24]. Through simulated cases, trainees can learn to weigh different options and communicate their decisions effectively to their colleagues and patients. AI can monitor the performance of trainees during simulated training, where it may be difficult for a trainer to observe and provide feedback in real-time. Additionally, AI can provide real-time feedback and coaching on non-technical skills such as teamwork and leadership, helping trainees to develop the attitudes and behaviors that are essential for success in the surgical field. As no modern surgeon works alone, surgeons must be trained to work within and with a team already at an early stage. By working with AI-powered virtual patients and scenarios, trainees can learn to navigate the social and emotional aspects of surgery and develop the empathy and professionalism that are crucial for building trust and rapport with patients [25–27]. This is of special importance, as training for adverse events is lacking, and surgeons have traditionally been less familiar with the training of

non-technical skills [23,28]. Further, using AI for training such situations can be particularly useful where real-life training is not feasible. Datasets from past adverse events can be analyzed by AI to help identify patterns or trends that may help inform training strategies. Thus, instructors can tailor their training to address situations that are most likely to occur, or to identify areas where additional training is needed. By tailoring training to address the specific needs of the trainee, practice becomes more efficient, allowing adequate time for other activities in the time-pressed life of surgical learners.

While AI shows utility and clear future potential within surgical learning, it also poses new challenges to instructors, who are forced to learn new competencies in order to stay current in medical education. Contemporary surgical care often involves a team approach, with surgeons collaborating with various professionals to provide comprehensive patient care. As a result, surgical training is shifting to be more collaborative and interdisciplinary, focusing on developing communication skills. While simulations can help teach surgical knowledge, technical skills, and non-technical aspects, the versatility of AI will be essential for rearing the coming generations of surgeons who are no longer pure masters of technical proficiency. Nevertheless, as discussed earlier, while the assessment of technical tasks is straightforward, the assessment of more varied simulations may pose challenges.

### 3.1.2. AI in Surgical Diagnostics and Decision-Making

As healthcare evolves, so does diagnostics and decision-making, placing the surgeon-in-training in new circumstances. AI has the potential to significantly improve the accuracy and efficiency of surgical diagnosis. Machine learning is a field of AI that is used to identify complex relationships from dependent and independent variables. Machine learning algorithms can analyze large amounts of data from various sources, such as medical images, laboratory test results, and patient history, to identify patterns and make predictions about the most likely diagnosis [29,30]. Machine learning algorithms can be trained on large datasets to recognize patterns and trends, helping clinicians to make more accurate and reliable diagnoses. These algorithms have been reported to provide comparable results to clinicians, especially within radiology [31–33]. These tools can be particularly useful in situations where there is a high volume of images to review or when there is a shortage of trained human analysts. Similarly, machine learning algorithms have shown utility in other specialties, such as dermatology and pathology. In addition, AI can assist in the development of diagnostic decision support systems, which can provide recommendations for further testing or treatment based on the most current evidence and best practices. These systems can help to streamline the diagnostic process, reducing the time and resources required to reach a diagnosis and enabling faster and more effective treatment.

### 3.1.3. AI in in Learning Minimally Invasive Surgery

While minimally invasive surgery is not the main focus of this review, it is imperative for surgical learners to understand the current prospects of AI in minimally invasive surgery. AI has the potential to significantly improve the safety and effectiveness of minimally invasive and robotic surgery. AI can assist in the development of advanced navigation and guidance systems, which can help to improve the accuracy and precision of the surgical procedure. AI-powered image analysis tools can be used to identify and track surgical instruments and structures within the body, providing real-time guidance to the surgeon during the procedure [34]. These tools can also assist in the identification of unexpected intraoperative events or complications, allowing the surgeon to take appropriate action in a timely manner. AI can also assist in the development of machine learning algorithms that can analyze data from past surgeries to identify patterns and trends, allowing for more personalized and evidence-based approaches to surgical planning and execution. The algorithms can be trained on large datasets to recognize patterns and trends, allowing for more accurate and reliable surgical decision-making. Despite the potential of AI in minimally invasive surgery, the current literature is still lagging regarding the utility of AI in robotic surgery [35]. All surgical procedures should be recorded for many reasons, not the

least educational. Through self-assessment, surgeons improve in many ways; by recording the procedures, surgeons can draw benefits from both self-regulated and teacher-guided video-based learning to improve proficiency. The recorded performances can be analyzed by AI-based methods. However, the vast amounts of data generated during each operation poses new challenges for analysis [35–37]. In addition, while learning minimally invasive surgery entails new modes of learning different from traditional surgery, the advent and implementation of AI-based technologies presents new challenges for surgeons of all ages. The demand for comprehensive data collection also dictates new ways for the recording of operations. Surgical approaches, where visualization is enhanced through optical systems, are especially suited for applications of AI such as robotic surgery [38]. Current daVinci systems enhance optics by the use of injected indocyanine green fluorescent dyes, which help the surgeon visualize blood vessels. Future robot versions will most likely incorporate various fluorophobes for real-time visualization of tissues to be manipulated, adding a further degree of precision.

### 3.1.4. Limitations of AI in Medical Education

Although AI entails numerous benefits and improves medical education, it is not without limitations (Table 1). While AI changes the role of doctors as medical communicators [39], it is feared to objectify patients and may be liable to miss subtle nuances in patient–doctor communication. Further, AI is far from ready for independent operation, requiring lifelong guidance for proper medical application [40]. AI may also currently lack the ability to detect conversational cues, which may help in guiding communication to individual levels suited for best delivering the information to each patient in a personalized way [6]. Artificial intelligence needs large amounts of data for machine-learning to enable robust decision-making. In some surgical specialties, the number of specific diagnoses is limited, hampering the application of machine-learning in given situations. For machine learning algorithms to learn and improve their performance, they require large amounts of data. It can be difficult to collect and label these data accurately, possibly impacting the performance of the AI system. The current medical literature regarding AI is complicated by jargon unfamiliar to many clinicians [41], possibly limiting both adaptation and clinical implementation. Further, the literature surrounding AI and respective applications has expanded beyond expectation in recent years. Many reports mostly describe the potential of AI, but few show evidence-based utility for clinical applications. Clearly, systematic literature reviews are needed to summarize the collective evidence for current use of AI.

**Table 1.** Limitations of AI in medical education.

| Limitation | Consequences |
| --- | --- |
| Lack of human judgment | Decisions are based on data, rules, and prior experience, but lacks ability to understand context and nuances |
| Lack of domain expertise | AI systems may lack deep knowledge and experience, risk of incorrect diagnoses and treatment plans |
| Bias in data | AI systems rely on the data they are trained on; if data are limited, decisions have weak background and may be biased |
| Need for interpretability | The decisions made by AI systems may be difficult to interpret and thus trust |
| High cost | AI systems are expensive to develop, implement, and possibly also to maintain |

There are ethical considerations to consider when using AI in surgical learning. There is a potential for bias in the data used for training the system, and the potential for the AI system to make decisions opposite to that of the surgeon or patient. Therefore, while AI can assist the surgeon in learning and improving, surgeons must be strong within the knowledge component and maintain decisive roles within patient care. Further, the surgeon may find it challenging to understand how the AI system arrived at a particular

decision or recommendation. In addition, it is central that the AI systems are highly reliable and safe, as any errors could have catastrophic consequences, posing challenges when the AI system is learning and adapting. As the readiness of an AI system using machine learning depends on the specific application and the needs of the user, the implementation needs to be gradual and individual. There are several factors to be considered when determining readiness, of which the most important are safety, robustness, explainability, and ethical considerations.

## 4. Discussion of Future Prospects

It remains impossible to predict the ways in which technology will enhance surgical learning. The integration of AI technology is here to stay and holds numerous possibilities for future developments. Within surgical learning, AI can facilitate personalized and adaptive learning experiences for surgeons of all levels. Machine learning algorithms can be applied to analyze performance data to identify areas of improvement and strengths, not only to help the individual, but also the surgical community. AI can also be used for the creation of more realistic simulation environments, thereby enabling practice in controlled realistic immersive environments.

AI will augment all aspects of patient care which need to be taken into consideration in surgical training [36]. Preoperative evaluations will include a multitude of data to be processed by AI for both preoperative planning, intraoperative approaches, and prediction of postoperative complications [42–45]. Intraoperatively, AI seems particularly beneficial in anesthetic evaluations to detect aberrations prior to clinical manifestations [46–49], but also displays future utility for surgical applications regarding detection, classification, and visualization of anatomic structures [50–52]. Further, AI-based algorithms have proven useful in the prediction of bleeding risk assessment both intraoperatively and postoperatively [53–55]. In addition, AI can be applied for postoperative evaluation of pain [56,57], diagnostics, and in the planning of postoperative care [58–61].

Intelligent tutoring systems can benefit from AI, which could provide real-time feedback and coaching during simulations [62–65]. These systems could analyze performance data, including instrument movement and tissue manipulation to provide objective and comprehensive evaluations of skill development and the attitude component of surgical competence [66–68]. Additionally, AI can be applied for diagnostic and treatment decision support system development, which could help in providing recommendations for best practices, thus helping them stay current. These systems could help streamline the diagnostic and treatment process, reducing the time and resources required to reach diagnoses, and reducing resource investment [69,70].

Future robots will improve upon current versions in many ways. Robots will need to be smaller in the future to enable wider applicability [71]. This especially applies for single-port surgery, but also for natural orifice surgery. In addition, microbots have been operated through external electromagnetic fields, enabling untethered use within the vitreal space [72]. Further, current surgical robots are limited in haptic feedback [71]. Robotic surgeons achieve haptics by visual cues; future robots will improve this area markedly in comparison to the current generation of robots. Enhanced visual feedback including the use of fluorophobes will improve in coming versions as well. AI-powered robots and other assistive technologies could in the future be applied to perform certain tasks during surgery. Such tasks could include technical aspects such as suturing, incisions, and closing of wounds, thus allowing the surgeon to focus on the essential components of surgery [73]. The role of the future surgeon will be different from what we are currently accustomed to. Nevertheless, fully autonomous surgical robots are not within the grasp to the current and coming generations of surgeons. Thus, the current role of surgical robots as telemanipulator systems operated by trained surgeons remains [71,73]. The surgeon can control the surgical robot from afar [74].

While telemedicine is tempting for robotic surgery, it can also be used for more basic healthcare, where machine learning and AI can help in delivering patient care to challenged

locations [75]. The recent shift to distant teaching due to the COVID-19 pandemic has paved way to further elaboration of surgical teaching [76]. It remains important to develop effective AI-based remote learning platforms for future situations [77]. The remote learning platforms enable both teacher-guided and self-regulated video-based learning. Through AI, more emphasis will be on teacher-guided learning with all the benefits of it included. One of the key advantages of using video-based learning in conjunction with AI is the ability to personalize the learning experience as mentioned earlier. With AI-based tools, the system can analyze student data and adjust the instructional content accordingly. Through this, each student is being presented with material that is tailored to their specific needs and abilities. Another advantage of using video-based learning with AI is the ability to provide interactive and engaging learning experiences. In addition, AI-based tools can provide real-time feedback to students, which can help them identify areas where they need further study. It seems evident that more and more teaching is done electronically with the aid of AI in the future. It is important to exercise caution when implementing self-regulated or AI-guided video-based learning, as it should be viewed as a supplement to, rather than a replacement for, traditional in-person teaching. While video-based learning with AI can be a powerful tool for improving student learning outcomes, it is not a substitute for the personal interaction, guidance, and support that students receive from their teachers. Therefore, it is important to ensure that these technologies are used in a way that complements, rather than replaces, traditional teaching methods. Teacher guidance is crucial, as the teacher is able to help the student to understand the video content in the context of their own learning and apply it to the curriculum. Teachers can also provide immediate feedback, propose thought-provoking scenarios, and use their own experiences to help the students put the video content in context to real-life clinical scenarios. It also remains essential to understand that the use of video-based learning with AI is not an omnipotent solution for all students. Therefore, it is important to conduct further studies to evaluate and refine the approach and engage clinical teachers and students in the decision-making process when deciding on the use of video-based learning with AI to ensure that this approach is effective and meets the needs and demands of the specific topic to be addressed. It is crucial to approach this integration to regular teaching with a diligent, thoughtful, evidence-based, and iterative approach in order to deliver effective future surgical education.

Ensuring adequate competency is essential for proper surgical training. While most previous studies on surgical learning have used objective assessments, there is a need to clearly define and measure competency, especially regarding the implementation of AI in surgical learning. This also applies to the transition from technical and virtual to clinical proficiency. While AI can be used to train knowledge and skills exceptionally well, surgical learners must have the right attitude and willingness to learn. AI can detect attitudes and willingness; however, those qualities and their grooming remains an individual task for students themselves. Modern surgical curricula development should embrace AI as a resource with vast opportunities.

In conclusion, the use of AI in surgical learning has the potential to significantly improve patient care by enhancing the efficiency and effectiveness of surgical training. There is a need for more systematic analysis of the current literature on AI in the field of surgical learning, and it is important to ensure that large datasets are used in a way that provides context for interpretation and considers the clinical implications. While machine learning may not yet be at the same level as human learning, it can be used to augment our cognitive abilities and allow us to focus on more complex and creative tasks. It is important to use AI appropriately and not become overly reliant on it, as it should be viewed as a tool to enhance rather than replace human learning.

**Author Contributions:** Conceptualization, N.P. and S.A.; methodology, N.P.; software, N.P.; validation, N.P., S.A. and T.L.; formal analysis, N.P.; investigation, N.P.; resources, N.P.; data curation, N.P.; writing—original draft preparation, N.P.; writing—review and editing, N.P., S.A. and T.L.; supervision, N.P.; project administration, N.P. All authors have read and agreed to the published version of the manuscript.

**Funding:** This research received no external funding.

**Institutional Review Board Statement:** Not applicable.

**Informed Consent Statement:** Not applicable.

**Data Availability Statement:** Data sharing not applicable.

**Conflicts of Interest:** The authors declare no conflict of interest.

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
