# Peer review of "Artificial Intelligence in Surgical Learning"

_2673-4095, doi:10.3390/surgeries4010010_

Round 1

Reviewer 1 Report

Many thanks for the opportunity to review the manuscript: “Artificial intelligence in surgical learning”. The article is a scoping review describing modern applications of AI to surgical training. I make the following suggestions:

  • I suggest you introduce the concept of artificial intelligence and its subdivisions (machine learning, computer vision etc) in the introduction. 
  • Methodology section is lacking in detail. Was there any human screening of titles or abstracts, or was this all performed by GPT3? Further, the search strategy seems very narrow. Can you be confident you have picked up all relevant papers? Note - this is not essential, given this is a scoping review, but it is appropriate to be thorough. 
  • The paper describes numerous examples of where AI could impact surgical training, but does not explain how this improvement would occur. E.g. “In addition 119 to improving technical skills, simulation-based training with AI can also help surgeons 120 develop non-technical skills such as crisis management and decision-making under 121 pressure.” There are no specifics here. 
  • Figure 2 is difficult to read. 

Author Response

Many thanks for the opportunity to review the manuscript: “Artificial intelligence in surgical learning”. The article is a scoping review describing modern applications of AI to surgical training.I make the following suggestions:

  • I suggest you introduce the concept of artificial intelligence and its subdivisions (machine learning, computer vision etc) in the introduction.

Author’s response: We thank for the positive assessment and the reviewer’s efforts. We have now included an introduction of artificial intelligence and its subdivisions accordingly. This is highlighted in yellow in the manuscript, lines 23-27 and stands as follows:

“AI refers to the simulation of human intelligence in computers, where AI systems can perform tasks that require human-like intelligence like speech recognition, visual perception, pattern-recognition, decision-making, and language processing. AI has several subdivisions, including machine learning, natural language processing, computer vision, and robotics.”

  • Methodology section is lacking in detail. Was there any human screening of titles or abstracts, or was this all performed by GPT3? Further, the search strategy seems very narrow. Can you be confident you have picked up all relevant papers? Note - this is not essential, given this is a scoping review, but it is appropriate to be thorough.

Author’s response: We thank the reviewer for this specification. Title, abstract and text-screenings were performed by the authors. GPT3 was used for summarizing some of the texts. We acknowledge the narrow search strategy; however, this was a conscious decision due to the format of the review. The subject definitely requires a systematic review to be performed at later stage. 

We have amended the Methodology as follows (lines 86-89): “Data was extracted manually in several increments, starting with title and abstract scanning, proceeding to text review. AI (OpenAI GPT3) was applied for literature analysis in select situations for summarizing data from the included texts.”

  • The paper describes numerous examples of where AI could impact surgical training, but does not explain how this improvement would occur. E.g. “In addition 119 to improving technical skills, simulation-based training with AI can also help surgeons 120 develop non-technical skills such as crisis management and decision-making under 121 pressure.” There are no specifics here.

Author’s response: We thank the reviewer for this comment. We imply that through the application of AI, surgical training can adapt a competence-framed approach with all components of surgical competency included. A comprehensive approach is occasionally lacking in modern surgical training. We would like to refer to a recently submitted systematic review from our group on how surgeons actually learn, but unfortunately it is still under review.

  • Figure 2 is difficult to read.

Author’s response: We acknowledge this problem. We think that it may be related to technical problems of replicating the figure in the manuscript. We kindly ask the editorial office for technical assistance in optimising the presentation of the image accordingly.

Reviewer 2 Report

Dear authors,

I read with great interest your article entitled "Artificial Intelligence in surgical learning". I believe that the purpose of this Review is extremely intelligent and useful for many aspects within the scientific community. AI is evolving more and more and becoming also a part of clinical and surgical practice. the role and the knowledge of AI in surgical learning is a fundamental aspect that should be further implemented.

One of the strengths of this article is the careful research of literature, which I think is exhaustive. However, I would like to suggest some changes in order to easily improve the manuscript:

-I think the purpose of the article is adequately explained in the background section of the abstract, however I would recommend excluding the phrase "however, clinical application is lagging" from the Results section, as it is repetitive and already mentioned earlier in the abstract.

-I have repeatedly read the words "rapid review" (for example in line 9 and line 80). I believe that this word doesn’t fit well with your work. A review should be a work of high value for the scientific community and although it is not a systematic review, it should be indicated with adjectives that are more appropriate with your elegant manuscript.

-In section 3 "results of current status of AI in surgical learning" are mentioned several reasons why AI can help in surgical learning. I think that it might be interesting, as well as more schematic, to insert a summary table containing the different categories of AI applications and the associated articles (Example on the main categories of AI applications: AI in learning surgical competence, AI in surgical diagnostics and decision making, AI in learning minimally invasive surgery). Within the main categories of AI applications, any subcategories you mentioned may also be added in the table (Example of subcategory for AI in learning surgical competence: Simulation based training, Clinical decision making, Data extraction and analysis, etc.). The purpose is to present an overview of the present utilization of AI in surgical learning, as you thought to do with figure number 2. However the figure number 2 turns out to be poor quality and in some places unreadable. Though interesting, this problem would inevitably lead the reader not to analyze the figure.

-Also in the 3.1.4 section “Limitations of AI in medical education”, could be interesting to insert a resumptive

Table of the principals limitation points of AI.

- I think the discussion paragraph is comprehensive. However, I think it might be useful to include a Conclusion Session, which could start with Row number 364.

Author Response

Dear authors,

I read with great interest your article entitled "Artificial Intelligence in surgical learning". I believe that the purpose of this Review is extremely intelligent and useful for many aspects within the scientific community. AI is evolving more and more and becoming also a part of clinical and surgical practice. the role and the knowledge of AI in surgical learning is a fundamental aspect that should be further implemented.

One of the strengths of this article is the careful research of literature, which I think is exhaustive. However, I would like to suggest some changes in order to easily improve the manuscript:

-I think the purpose of the article is adequately explained in the background section of the abstract, however I would recommend excluding the phrase "however, clinical application is lagging" from the Results section, as it is repetitive and already mentioned earlier in the abstract.

Author’s response: We thank for the positive assessment and the reviewer’s efforts. We have now excluded the phrase from the Results accordingly (line 12): “AI presents with great potential within robotic surgery specifically; however, clinical application is lagging;”

-I have repeatedly read the words "rapid review" (for example in line 9 and line 80). I believe that this word doesn’t fit well with your work. A review should be a work of high value for the scientific community and although it is not a systematic review, it should be indicated with adjectives that are more appropriate with your elegant manuscript.

Author’s response: Thank you for the appropriate comment. We agree completely and have modified the statement accordingly by removing the word “rapid” from both instances (marked with yellow highlight and strikethrough).

-In section 3 "results of current status of AI in surgical learning" are mentioned several reasons why AI can help in surgical learning. I think that it might be interesting, as well as more schematic, to insert a summary table containing the different categories of AI applications and the associated articles (Example on the main categories of AI applications: AI in learning surgical competence, AI in surgical diagnostics and decision making, AI in learning minimally invasive surgery). Within the main categories of AI applications, any subcategories you mentioned may also be added in the table (Example of subcategory for AI in learning surgical competence: Simulation based training, Clinical decision making, Data extraction and analysis, etc.). The purpose is to present an overview of the present utilization of AI in surgical learning, as you thought to do with figure number 2. However the figure number 2 turns out to be poor quality and in some places unreadable. Though interesting, this problem would inevitably lead the reader not to analyze the figure.

Author’s response: Thank you for the appropriate comment. We acknowledge the problem with figure 2. We think that it may be related to technical problems of replicating the figure in the manuscript. We kindly ask the editorial office for technical assistance in optimising the presentation of the image accordingly.

-Also in the 3.1.4 section “Limitations of AI in medical education”, could be interesting to insert a resumptive Table of the principals limitation points of AI.

Author’s response: We thank the reviewer for this comment. We have now amended the manuscript with a table addressing the limitations of AI in medical education lines (278-279).

- I think the discussion paragraph is comprehensive. However, I think it might be useful to include a Conclusion Session, which could start with Row number 364.

Author’s response:  We thank the reviewer for the positive comments. We kindly refer to the paragraph starting on line 369 which is meant to form a Conclusion of the findings.